# Column Selection via Adaptive Sampling

**Saurabh Paul**
Global Risk Sciences, Paypal Inc.
saupaul@paypal.com

**Malik Magdon-Ismail**
CS Dept., Rensselaer Polytechnic Institute
magdon@cs.rpi.edu

**Petros Drineas**
CS Dept., Rensselaer Polytechnic Institute
drinep@cs.rpi.edu

## Abstract

Selecting a good column (or row) subset of massive data matrices has found many applications in data analysis and machine learning. We propose a new adaptive sampling algorithm that can be used to improve *any* relative-error column selection algorithm. Our algorithm delivers a tighter theoretical bound on the approximation error which we also demonstrate empirically using two well known relative-error column subset selection algorithms. Our experimental results on synthetic and real-world data show that our algorithm outperforms non-adaptive sampling as well as prior adaptive sampling approaches.

## 1 Introduction

In numerous machine learning and data analysis applications, the input data are modelled as a matrix $\mathbf{A} \in \mathbb{R}^{m \times n}$, where $m$ is the number of objects (data points) and $n$ is the number of features. Often, it is desirable to represent your solution using a few features (to promote better generalization and interpretability of the solutions), or using a few data points (to identify important coresets of the data), for example PCA, sparse PCA, sparse regression, coreset based regression, etc. [1, 2, 3, 4]. These problems can be reduced to identifying a good subset of the columns (or rows) in the data matrix, the column subset selection problem (CSSP). For example, finding an optimal sparse linear encoder for the data (dimension reduction) can be explicitly reduced to CSSP [5]. Motivated by the fact that in many practical applications, the left and right singular vectors of a matrix $\mathbf{A}$ lacks any physical interpretation, a long line of work [6, 7, 8, 9, 10, 11, 12, 13, 14, 15], focused on extracting a subset of columns of the matrix $\mathbf{A}$, which are approximately as good as $\mathbf{A}_k$ at reconstructing $\mathbf{A}$. To make our discussion more concrete, let us formally define CSSP.

**Column Subset Selection Problem, CSSP:** Find a matrix $\mathbf{C} \in \mathbb{R}^{m \times c}$ containing $c$ columns of $\mathbf{A}$ for which $\left\| \mathbf{A} - \mathbf{C}\mathbf{C}^+\mathbf{A} \right\|_F$ is small.[1] In the prior work, one measures the quality of a CSSP-solution against $\mathbf{A}_k$, the best rank-$k$ approximation to $\mathbf{A}$ obtained via the singular value decomposition (SVD), where $k$ is a user specified target rank parameter. For example, [15] gives efficient algorithms to find $\mathbf{C}$ with $c \approx 2k/\epsilon$ columns, for which $\left\| \mathbf{A} - \mathbf{C}\mathbf{C}^+\mathbf{A} \right\|_F \leq (1 + \epsilon) \left\| \mathbf{A} - \mathbf{A}_k \right\|_F$.

Our contribution is not to directly attack CSSP. We present a novel algorithm that can improve an existing CSSP algorithm by adaptively invoking it, in a sense *actively* learning which columns to sample next based on the columns you have already sampled. If you use the CSSP-algorithm from [15] as a strawman benchmark, you can obtain $c$ columns all at once and incur an error roughly $(1 + 2k/c) \left\| \mathbf{A} - \mathbf{A}_k \right\|_F$. Or, you can invoke the algorithm to obtain, for example, $c/2$ columns, and then allow the algorithm to *adapt* to the columns already chosen (for example by modifying $\mathbf{A}$) before choosing the remaining $c/2$ columns. We refer to the former as continued sampling and to the

latter as adaptive sampling. We prove performance guarantees which show that adaptive sampling improves upon continued sampling, and we present experiments on synthetic and real data that demonstrate significant empirical performance gains.

## 1.1 Notation

$\mathbf{A}, \mathbf{B}, \ldots$ denote matrices and $\mathbf{a}, \mathbf{b}, \ldots$ denote column vectors; $\mathbf{I}_n$ is the $n \times n$ identity matrix. $[\mathbf{A}, \mathbf{B}]$ and $[\mathbf{A}; \mathbf{B}]$ denote matrix concatenation operations in a column-wise and row-wise manner, respectively. Given a set $S \subseteq \{1, \ldots n\}$, $\mathbf{A}_S$ is the matrix that contains the columns of $\mathbf{A} \in \mathbb{R}^{m \times n}$ indexed by $S$. Let $\text{rank}(\mathbf{A}) = \rho \leq \min\{m, n\}$. The (economy) SVD of $\mathbf{A}$ is

$$\mathbf{A} = (\mathbf{U}_k \ \mathbf{U}_{\rho-k}) \begin{pmatrix} \mathbf{\Sigma}_k & \mathbf{0} \\ \mathbf{0} & \mathbf{\Sigma}_{\rho-k} \end{pmatrix} \begin{pmatrix} \mathbf{V}_k^T \\ \mathbf{V}_{\rho-k}^T \end{pmatrix} = \sum_{i=1}^{\rho} \sigma_i(\mathbf{A}) \mathbf{u}_i \mathbf{v}_i^T$$

where $\mathbf{U}_k \in \mathbb{R}^{m \times k}$ and $\mathbf{U}_{\rho-k} \in \mathbb{R}^{m \times (\rho-k)}$ contain the left singular vectors $\mathbf{u}_i$, $\mathbf{V}_k \in \mathbb{R}^{n \times k}$ and $\mathbf{V}_{\rho-k} \in \mathbb{R}^{n \times (\rho-k)}$ contain the right singular vectors $\mathbf{v}_i$, and $\mathbf{\Sigma} \in \mathbb{R}^{\rho \times \rho}$ is a diagonal matrix containing the singular values $\sigma_1(\mathbf{A}) \geq \ldots \geq \sigma_\rho(\mathbf{A}) > 0$. The Frobenius norm of $\mathbf{A}$ is $\|\mathbf{A}\|_F^2 = \sum_{i,j} \mathbf{A}_{ij}^2$; $\text{Tr}(\mathbf{A})$ is the trace of $\mathbf{A}$; the pseudoinverse of $\mathbf{A}$ is $\mathbf{A}^+ = \mathbf{V}\mathbf{\Sigma}^{-1}\mathbf{U}^T$; and, $\mathbf{A}_k$, the best rank-$k$ approximation to $\mathbf{A}$ under any unitarily invariant norm is $\mathbf{A}_k = \mathbf{U}_k\mathbf{\Sigma}_k\mathbf{V}_k^T = \sum_{i=1}^{k} \sigma_i \mathbf{u}_i \mathbf{v}_i^T$.

## 1.2 Our Contribution: Adaptive Sampling

We design a novel CSSP-algorithm that *adaptively* selects columns from the matrix $\mathbf{A}$ in *rounds*. In each round we *remove* from $\mathbf{A}$ the information that has already been "captured" by the columns that have been thus far selected. Algorithm 1 selects $tc$ columns of $\mathbf{A}$ in $t$ rounds, where in each round $c$ columns of $\mathbf{A}$ are selected using a relative-error CSSP-algorithm from prior work.

---

**Input:** $\mathbf{A} \in \mathbb{R}^{m \times n}$; target rank $k$; # rounds $t$; columns per round $c$
**Output:** $\mathbf{C} \in \mathbb{R}^{m \times tc}$, $tc$ columns of $\mathbf{A}$ and $S$, the indices of those columns.
  1: $S = \{\}$; $\mathbf{E}^0 = \mathbf{A}$
  2: **for** $\ell = 1, \cdots, t$ **do**
  3:     Sample indices $S_\ell$ of $c$ columns from $\mathbf{E}^{\ell-1}$ using a CSSP-algorithm.
  4:     $S \leftarrow S \cup S_\ell$.
  5:     Set $\mathbf{C} = \mathbf{A}_S$ and $\mathbf{E}^\ell = \mathbf{A} - (\mathbf{C}\mathbf{C}^+\mathbf{A})_{\ell k}$.
  6: **return** $\mathbf{C}, S$

---

**Algorithm 1:** Adaptive Sampling

At round $\ell$ in Step 3, we compute column indices $S$ (and $\mathbf{C} = \mathbf{A}_S$) using a CSSP-algorithm on the residual $\mathbf{E}^{\ell-1}$ of the previous round. To compute this residual, remove from $\mathbf{A}$ the best rank-$(\ell-1)k$ approximation to $\mathbf{A}$ in the span of the columns selected from the first $\ell-1$ rounds,

$$\mathbf{E}^{\ell-1} = \mathbf{A} - (\mathbf{C}\mathbf{C}^+\mathbf{A})_{(\ell-1)k}.$$

A similar strategy was developed in [8] with sequential adaptive use of (additive error) CSSP-algorithms. These (additive error) CSSP-algorithms select columns according to column norms [11]. In [8], the residual in step 5 is defined differently, as $\mathbf{E}^\ell = \mathbf{A} - \mathbf{C}\mathbf{C}^+\mathbf{A}$. To motivate our result, it helps to take a closer look at the reconstruction error $\mathbf{E} = \mathbf{A} - \mathbf{C}\mathbf{C}^+\mathbf{A}$ after $t$ adaptive rounds of the strategy in [8] with the CSSP-algorithm in [11].

| # rounds | **Continued sampling:** $tc$ columns using CSSP-algorithm from [11]. ($\epsilon = k/c$) | **Adaptive sampling:** $t$ rounds of the strategy in [8] with the CSSP-algorithm from [11]. |
|---|---|---|
| $t = 2$ | $\|\mathbf{E}\|_F^2 \leq \|\mathbf{A} - \mathbf{A}_k\|_F^2 + \dfrac{\epsilon}{2}\|\mathbf{A}\|_F^2$ | $\|\mathbf{E}\|_F^2 \leq (1+\epsilon)\|\mathbf{A} - \mathbf{A}_k\|_F^2 + \epsilon^2\|\mathbf{A}\|_F^2$ |
| $t$ | $\|\mathbf{E}\|_F^2 \leq \|\mathbf{A} - \mathbf{A}_k\|_F^2 + \dfrac{\epsilon}{t}\|\mathbf{A}\|_F^2$ | $\|\mathbf{E}\|_F^2 \leq (1+O(\epsilon))\|\mathbf{A} - \mathbf{A}_k\|_F^2 + \epsilon^t\|\mathbf{A}\|_F^2$ |

Typically $\|\mathbf{A}\|_F^2 \gg \|\mathbf{A} - \mathbf{A}_k\|_F^2$ and $\epsilon$ is small (i.e., $c \gg k$), so adaptive sampling à la [8] wins over continued sampling for additive error CSSP-algorithms. This is especially apparent after $t$ rounds, where continued sampling only attenuates the big term $\|\mathbf{A}\|_F^2$ by $\epsilon/t$, but adaptive sampling exponentially attenuates this term by $\epsilon^t$.

Recently, powerful CSSP-algorithms have been developed which give relative-error guarantees [15]. We can use the adaptive strategy from [8] together with these newer relative error CSSP-algorithms. If one carries out the analysis from [8] by replacing the additive error CSSP-algorithm from [11] with the relative error CSSP-algorithm in [15], the comparison of continued and adaptive sampling using the strategy from [8] becomes ($t = 2$ rounds suffices to see the problem):

| # rounds | **Continued sampling:** $tc$ columns using CSSP-algorithm from [15]. ($\epsilon = 2k/c$) | **Adaptive sampling:** $t$ rounds of the strategy in [8] with the CSSP-algorithm from [15]. |
|---|---|---|
| $t = 2$ | $\|\mathbf{E}\|_F^2 \leq \left(1 + \frac{\epsilon}{2}\right)\|\mathbf{A} - \mathbf{A}_k\|_F^2$ | $\|\mathbf{E}\|_F^2 \leq \left(1 + \frac{\epsilon}{2} + \frac{\epsilon^2}{2}\right)\|\mathbf{A} - \mathbf{A}_k\|_F^2$ |

Adaptive sampling from [8] gives a worse theoretical guarantee than continued sampling for relative error CSSP-algorithms. In a nutshell, no matter how many rounds of adaptive sampling you do, the theoretical bound will not be better than $(1 + k/c)\|\mathbf{A} - \mathbf{A}_k\|_F^2$ if you are using a relative error CSSP-algorithm. This raises an obvious question: is it possible to combine relative-error CSSP-algorithms with adaptive sampling to get (provably and empirically) improved CSSP-algorithms? The approach of [8] does not achieve this objective. We provide a positive answer to this question.

Our approach is a subtle modification to the approach in [8]: in Step 5 of Algorithm 1. When we compute the residual matrix in round $\ell$, we subtract $(\mathbf{C}\mathbf{C}^+\mathbf{A})_{\ell k}$ from $\mathbf{A}$, the best rank-$\ell k$ approximation to the projection of $\mathbf{A}$ onto the current columns selected, as opposed to subtracting the full projection $\mathbf{C}\mathbf{C}^+\mathbf{A}$. This subtle change, is critical in our new analysis which gives a tighter bound on the final error, allowing us to boost relative-error CSSP-algorithms. For $t = 2$ rounds of adaptive sampling, we get a reconstruction error of

$$\|\mathbf{E}\|_F^2 \leq (1 + \epsilon)\|\mathbf{A} - \mathbf{A}_{2k}\|_F^2 + \epsilon(1 + \epsilon)\|\mathbf{A} - \mathbf{A}_k\|_F^2,$$

where $\epsilon = 2k/c$. The critical improvement in the bound is that the dominant $O(1)$-term depends on $\|\mathbf{A} - \mathbf{A}_{2k}\|_F^2$, and the dependence on $\|\mathbf{A} - \mathbf{A}_k\|_F^2$ is now $O(\epsilon)$. To highlight this improved theoretical bound in an extreme case, consider a matrix $\mathbf{A}$ that has rank exactly $2k$, then $\|\mathbf{A} - \mathbf{A}_{2k}\|_F = 0$. Continued sampling gives an error-bound $(1 + \frac{\epsilon}{2})\|\mathbf{A} - \mathbf{A}_k\|_F^2$, where as our adaptive sampling gives an error-bound $(\epsilon + \epsilon^2)\|\mathbf{A} - \mathbf{A}_k\|_F^2$, which is clearly better in this extreme case. In practice, data matrices have rapidly decaying singular values, so this extreme case is not far from reality (See Figure 1).

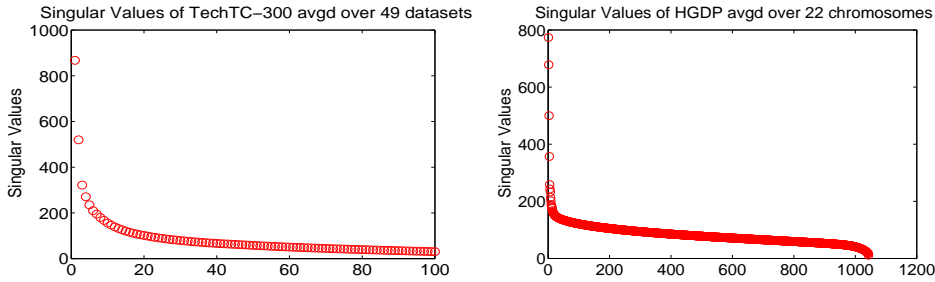

Figure 1: Figure showing the singular value decay for two real world datasets.

To state our main theoretical result, we need to more formally define a relative error CSSP-algorithm.

**Definition 1** (Relative Error CSSP-algorithm $\mathcal{A}(\mathbf{X}, k, c)$). *A relative error CSSP-algorithm $\mathcal{A}$ takes as input a matrix $\mathbf{X}$, a rank parameter $k < \operatorname{rank}(\mathbf{X})$ and a number of columns $c$, and outputs column indices $S$ with $|S| = c$, so that the columns $\mathbf{C} = \mathbf{X}_S$ satisfy:*

$$\mathbb{E}_\mathbf{C}\left[\|\mathbf{X} - (\mathbf{C}\mathbf{C}^+\mathbf{X})_k\|_F^2\right] \leq (1 + \epsilon(c, k))\|\mathbf{X} - \mathbf{X}_k\|_F^2,$$

*where $\epsilon(c, k)$ depends on $\mathcal{A}$ and the expectation is over random choices made in the algorithm.*[2]

Our main theorem bounds the reconstruction error when our adaptive sampling approach is used to boost $\mathcal{A}$. The boost in performance depends on the decay of the spectrum of $\mathbf{A}$.

**Theorem 1.** *Let $\mathbf{A} \in \mathbb{R}^{m \times n}$ be a matrix of rank $\rho$ and let $k < \rho$ be a target rank. If, in Step 3 of Algorithm 1, we use the relative error CSSP-algorithm $\mathcal{A}$ with $\epsilon(c, k) = \epsilon < 1$, then*

$$\mathbb{E}_{\mathbf{C}} \left[ \left\| \mathbf{A} - (\mathbf{C}\mathbf{C}^+\mathbf{A})_{tk} \right\|_F^2 \right] \leq (1 + \epsilon) \left\| \mathbf{A} - \mathbf{A}_{tk} \right\|_F^2 + \epsilon \sum_{i=1}^{t-1} (1 + \epsilon)^{t-i} \left\| \mathbf{A} - \mathbf{A}_{ik} \right\|_F^2 .$$

**Comments.**

1. The dominant $O(1)$ term in our bound is $\left\| \mathbf{A} - \mathbf{A}_{tk} \right\|_F$, not $\left\| \mathbf{A} - \mathbf{A}_k \right\|_F$. This is a major improvement since the former is typically *much smaller* than the latter in real data. Further, we need a bound on the reconstruction error $\left\| \mathbf{A} - \mathbf{C}\mathbf{C}^+\mathbf{A} \right\|_F$. Our theorem give a stronger result than needed because $\left\| \mathbf{A} - \mathbf{C}\mathbf{C}^+\mathbf{A} \right\|_F \leq \left\| \mathbf{A} - (\mathbf{C}\mathbf{C}^+\mathbf{A})_{tk} \right\|_F$.

2. We presented our result for the case of a relative error CSSP-algorithm with a guarantee on the expected reconstruction error. Clearly, if the CSSP-algorithm is deterministic, then Theorem 1 will also hold deterministically. The result in Theorem 1 can also be boosted to hold with high probability, by repeating the process $\log \frac{1}{\delta}$ times and picking the columns which performed best. Then, with probability at least $1 - \delta$,

$$\left\| \mathbf{A} - (\mathbf{C}\mathbf{C}^+\mathbf{A})_{tk} \right\|_F^2 \leq (1 + 2\epsilon) \left\| \mathbf{A} - \mathbf{A}_{tk} \right\|_F^2 + 2\epsilon \sum_{i=1}^{t-1} (1 + \epsilon)^{t-i} \left\| \mathbf{A} - \mathbf{A}_{ik} \right\|_F^2 .$$

   If the CSSP-algorithm itself only gives a high-probability (at least $1-\delta$) guarantee, then the bound in Theorem 1 also holds with high probability, at least $1 - t\delta$, which is obtained by applying a union bound to the probability of failure in each round.

3. Our results hold for any relative error CSSP-algorithm combined with our adaptive sampling strategy. The relative error CSSP-algorithm in [15] has $\epsilon(c, k) \approx 2k/c$. The relative error CSSP-algorithm in [16] has $\epsilon(c, k) = O(k \log k/c)$. Other algorithms can be found in [8, 9, 17]. We presented the simplest form of the result, which can be generalized to sample a different number of columns in each round, or even use a different CSSP-algorithm in each round. We have not optimized the sampling schedule, i.e. how many columns to sample in each round. At the moment, this is largely dictated by the CSSP algorithm itself, which requires a minimum number of samples in each round to give a theoretical guarantee. From the empirical perspective (for example using leverage score sampling to select columns), strongest performance may be obtained by adapting after every column is selected.

4. In the context of the additive error CSSP-algorithm from [11], our adaptive sampling strategy gives a theoretical performance guarantee which is at least as good as the adaptive sampling strategy from [8].

Lastly, we also provide the *first* empirical evaluation of adaptive sampling algorithms. We implemented our algorithm using two relative-error column selection algorithms (the near-optimal column selection algorithm of [18, 15] and the leverage-score sampling algorithm of [19]) and compared it against the adaptive sampling algorithm of [8] on synthetic and real-world data. The experimental results show that our algorithm outperforms prior approaches.

## 1.3 Related Work

Column selection algorithms have been extensively studied in prior literature. Such algorithms include rank-revealing QR factorizations [6, 20] for which only weak performance guarantees can be derived. The QR approach was improved in [21] where the authors proposed a memory efficient implementation. The randomized additive error CSSP-algorithm [11] was a breakthrough, which led to a series of improvements producing relative CSSP-algorithms using a variety of randomized and

deterministic techniques. These include leverage score sampling [19, 16], volume sampling [8, 9, 17], the two-stage hybrid sampling approach of [22], the near-optimal column selection algorithms of [18, 15], as well as deterministic variants presented in [23]. We refer the reader to Section 1.5 of [15] for a detailed overview of prior work. Our focus is not on CSSP-algorithms *per se*, but rather on adaptively invoking existing CSSP-algorithms. The only prior adaptive sampling with a provable guarantee was introduced in [8] and further analyzed in [24, 9, 25]; this strategy is specifically boosts the additive error CSSP-algorithm, but does not work with relative error CSSP-algorithms which are currently in use. Our modification of the approach in [8] is delicate, but crucial to the new analysis we perform in the context of relative error CSSP-algorithms.

Our work is motivated by relative error CSSP-algorithms satisfying definition 1. Such algorithms exist which give expected guarantees [15] as well as high probability guarantees [19]. Specifically, given $\mathbf{X} \in \mathbb{R}^{m \times n}$ and a target rank $k$, the leverage-score sampling approach of [19] selects $c = O\left((k/\epsilon^2) \log\left(k/\epsilon^2\right)\right)$ columns of $\mathbf{A}$ to form a matrix $\mathbf{C} \in \mathbb{R}^{m \times c}$ to give a $(1+\epsilon)$-relative error with probability at least $1 - \delta$. Similarly, [18, 15] proposed near-optimal relative error CSSP-algorithms selecting $c \approx 2c/\epsilon$ columns and giving a $(1 + \epsilon)$-relative error guarantee in expectation, which can be boosted to a high probability guarantee via independent repetition.

## 2 Proof of Theorem 1

We now prove the main result which analyzes the performance of our adaptive sampling in Algorithm 1 for a relative error CSSP-algorithm. We will need the following linear algebraic Lemma.

**Lemma 1.** *Let* $\mathbf{X}, \mathbf{Y} \in \mathbb{R}^{m \times n}$ *and suppose that* $\mathrm{rank}(\mathbf{Y}) = r$. *Then,*

$$\sigma_i(\mathbf{X} - \mathbf{Y}) \geq \sigma_{r+i}(\mathbf{X}).$$

*Proof.* Observe that $\sigma_i(\mathbf{X} - \mathbf{Y}) = \|(\mathbf{X} - \mathbf{Y}) - (\mathbf{X} - \mathbf{Y})_{i-1}\|_2$. The claim is now immediate from the Eckart-Young theorem because $\mathbf{Y} + (\mathbf{X} - \mathbf{Y})_{i-1}$ has rank at most $r + i - 1$, therefore

$$\sigma_i(\mathbf{X} - \mathbf{Y}) = \|\mathbf{X} - (\mathbf{Y} + (\mathbf{X} - \mathbf{Y})_{i-1})\|_2 \geq \|\mathbf{X} - \mathbf{X}_{r+i-1}\|_2 = \sigma_{r+i}(\mathbf{X}).$$

$\square$

We are now ready to prove Theorem 1 by induction on $t$, the number of rounds of adaptive sampling. When $t = 1$, the claim is that

$$\mathbb{E}\left[\|\mathbf{A} - (\mathbf{C}\mathbf{C}^+\mathbf{A})_k\|_F^2\right] \leq (1 + \epsilon) \|\mathbf{A} - \mathbf{A}_k\|_F^2,$$

which is immediate from the definition of the relative error CSSP-algorithm. Now for the induction. Suppose that after $t$ rounds, columns $\mathbf{C}^t$ are selected, and we have the induction hypothesis that

$$\mathbb{E}_{\mathbf{C}^t}\left[\|\mathbf{A} - (\mathbf{C}^t\mathbf{C}^{t+}\mathbf{A})_{tk}\|_F^2\right] \leq (1 + \epsilon) \|\mathbf{A} - \mathbf{A}_{tk}\|_F^2 + \epsilon \sum_{i=1}^{t-1}(1 + \epsilon)^{t-i} \|\mathbf{A} - \mathbf{A}_{ik}\|_F^2. \quad (1)$$

In the $(t + 1)$th round, we use the residual $\mathbf{E}^t = \mathbf{A} - (\mathbf{C}^t\mathbf{C}^{t+}\mathbf{A})_{tk}$ to select new columns $\mathbf{C}'$. Our relative error CSSP-algorithm $\mathcal{A}$ gives the following guarantee:

$$\begin{aligned}
\mathbb{E}_{\mathbf{C}'}\left[\|\mathbf{E}^t - (\mathbf{C}'\mathbf{C}'^+\mathbf{E}^t)_k\|_F^2 \,\Big|\, \mathbf{E}^t\right] &\leq (1 + \epsilon) \|\mathbf{E}^t - \mathbf{E}_k^t\|_F^2 \\
&= (1 + \epsilon)\left(\|\mathbf{E}^t\|_F^2 - \sum_{i=1}^{k}\sigma_i^2(\mathbf{E}^t)\right) \\
&\leq (1 + \epsilon)\left(\|\mathbf{E}^t\|_F^2 - \sum_{i=1}^{k}\sigma_{tk+i}^2(\mathbf{A})\right). \quad (2)
\end{aligned}$$

(The last step follows because $\sigma_i^2(\mathbf{E}^t) = \sigma_i^2(\mathbf{A} - (\mathbf{C}^t\mathbf{C}^{t+}\mathbf{A})_{tk})$ and we can apply Lemma 1 with $\mathbf{X} = \mathbf{A}$, $\mathbf{Y} = (\mathbf{C}^t\mathbf{C}^{t+}\mathbf{A})_{tk}$ and $r = \mathrm{rank}(\mathbf{Y}) = tk$, to obtain $\sigma_i^2(\mathbf{E}^t) \geq \sigma_{tk+i}^2(\mathbf{A})$.) We now take

the expectation of both sides with respect to the columns $\mathbf{C}^t$,

$$\mathbb{E}_{\mathbf{C}^t}\left[\mathbb{E}_{\mathbf{C}'}\left[\left\|\mathbf{E}^t - (\mathbf{C}'\mathbf{C}'^{+}\mathbf{E}^t)_k\right\|_F^2 \middle| \mathbf{E}^t\right]\right]$$

$$\leq (1+\epsilon)\left(\mathbb{E}_{\mathbf{C}^t}\left[\left\|\mathbf{E}^t\right\|_F^2\right] - \sum_{i=1}^{k}\sigma_{tk+i}^2(\mathbf{A})\right).$$

$$\overset{(a)}{\leq} (1+\epsilon)^2\|\mathbf{A}-\mathbf{A}_{tk}\|_F^2 + \epsilon\sum_{i=1}^{t-1}(1+\epsilon)^{t+1-i}\|\mathbf{A}-\mathbf{A}_{ik}\|_F^2 - (1+\epsilon)\sum_{i=1}^{k}\sigma_{tk+i}^2(\mathbf{A})$$

$$= (1+\epsilon)\left(\|\mathbf{A}-\mathbf{A}_{tk}\|_F^2 - \sum_{i=1}^{k}\sigma_{tk+i}^2(\mathbf{A})\right) + \epsilon(1+\epsilon)\|\mathbf{A}-\mathbf{A}_{tk}\|_F^2$$

$$+ \epsilon\sum_{i=1}^{t-1}(1+\epsilon)^{t+1-i}\|\mathbf{A}-\mathbf{A}_{ik}\|_F^2$$

$$= (1+\epsilon)\|\mathbf{A}-\mathbf{A}_{(t+1)k}\|_F^2 + \epsilon\sum_{i=1}^{t}(1+\epsilon)^{t+1-i}\|\mathbf{A}-\mathbf{A}_{ik}\|_F^2 \qquad (3)$$

(a) follows, because of the induction hypothesis (eqn. 1). The columns chosen after round $t+1$ are $\mathbf{C}^{t+1} = [\mathbf{C}^t, \mathbf{C}']$. By the law of iterated expectation,

$$\mathbb{E}_{\mathbf{C}^t}\left[\mathbb{E}_{\mathbf{C}'}\left[\left\|\mathbf{E}^t - (\mathbf{C}'\mathbf{C}'^{+}\mathbf{E}^t)_k\right\|_F^2 \middle| \mathbf{E}^t\right]\right] = \mathbb{E}_{\mathbf{C}^{t+1}}\left[\left\|\mathbf{E}^t - (\mathbf{C}'\mathbf{C}'^{+}\mathbf{E}^t)_k\right\|_F^2\right].$$

Observe that $\mathbf{E}^t - (\mathbf{C}'\mathbf{C}'^{+}\mathbf{E}^t)_k = \mathbf{A} - (\mathbf{C}^t\mathbf{C}^{t+}\mathbf{A})_{tk} - (\mathbf{C}'\mathbf{C}'^{+}\mathbf{E}^t)_k = \mathbf{A} - \mathbf{Y}$, where $\mathbf{Y}$ is in the column space of $\mathbf{C}^{t+1} = [\mathbf{C}^t, \mathbf{C}']$; further, $\text{rank}(\mathbf{Y}) \leq (t+1)k$. Since $(\mathbf{C}^{t+1}\mathbf{C}^{t+1^{+}}\mathbf{A})_{(t+1)k}$ is the best rank-$(t+1)k$ approximation to $\mathbf{A}$ in the column space of $\mathbf{C}^{t+1}$, for any realization of $\mathbf{C}^{t+1}$,

$$\left\|\mathbf{A} - (\mathbf{C}^{t+1}\mathbf{C}^{t+1^{+}}\mathbf{A})_{(t+1)k}\right\|_F^2 \leq \left\|\mathbf{E}^t - (\mathbf{C}'\mathbf{C}'^{+}\mathbf{E}^t)_k\right\|_F^2. \qquad (4)$$

Combining (4) with (3), we have that

$$\mathbb{E}_{\mathbf{C}^{t+1}}\left[\left\|\mathbf{A} - (\mathbf{C}^{t+1}\mathbf{C}^{t+1^{+}}\mathbf{A})_{(t+1)k}\right\|_F^2\right] \leq (1+\epsilon)\|\mathbf{A}-\mathbf{A}_{(t+1)k}\|_F^2 + \epsilon\sum_{i=1}^{t}(1+\epsilon)^{t+1-i}\|\mathbf{A}-\mathbf{A}_{ik}\|_F^2.$$

This is the desired bound after $t+1$ rounds, concluding the induction. $\qquad\square$

It is instructive to understand where our new adaptive sampling strategy is needed for the proof to go through. The crucial step is (2) where we use Lemma 1 – it is essential that the residual was a low-rank perturbation of $\mathbf{A}$.

## 3 Experiments

We compared three adaptive column sampling methods, using two real and two synthetic data sets.[3]

### Adaptive Sampling Methods

**ADP-AE:** the prior adaptive method which uses the additive error CSSP-algorithm [8].

**ADP-LVG:** our new adaptive method using the relative error CSSP-algorithm [19].

**ADP-Nopt:** our adaptive method using the near optimal relative error CSSP-algorithm [15].

### Data Sets

**HGDP 22 chromosomes:** SNPs human chromosome data from the HGDP database [26]. We use all 22 chromosome matrices (1043 rows; 7,334-37,493 columns) and report the average. Each matrix contains $+1, 0, -1$ entries, and we randomly filled in missing entries.

**TechTC-300:** 49 document-term matrices [27] (150-300 rows (documents); 10,000-40,000 columns (words)). We kept 5-letter or larger words and report averages over 49 data-sets.

**Synthetic 1:** Random $1000 \times 10000$ matrices with $\sigma_i = i^{-0.3}$ (power law).

**Synthetic 2:** Random $1000 \times 10000$ matrices with $\sigma_i = \exp^{(1-i)/10}$ (exponential).

For randomized algorithms, we repeat the experiments five times and take the average. We use the synthetic data sets to provide a controlled environment in which we can see performance for different types of singular value spectra on very large matrices. In prior work it is common to report on the quality of the columns selected $\mathbf{C}$ by comparing the best rank-$k$ approximation within the column-span of $\mathbf{C}$ to $\mathbf{A}_k$. Hence, we report the relative error $\left\|\mathbf{A} - (\mathbf{C}\mathbf{C}^+\mathbf{A})_k\right\|_F / \left\|\mathbf{A} - \mathbf{A}_k\right\|_F$ when comparing the algorithms. We set the target rank $k = 5$ and the number of columns in each round to $c = 2k$. We have tried several choices for $k$ and $c$ and the results are qualitatively identical so we only report on one choice. Our first set of results in Figure 2 is to compare the prior adaptive algorithm **ADP-AE** with the new adaptive ones **ADP-LVG** and **ADP-Nopt** which boose relative error CSSP-algorithms. Our two new algorithms are both performing better the prior existing adaptive sampling algorithm. Further, **ADP-Nopt** is performing better than **ADP-LVG**, and this is also not surprising, because **ADP-Nopt** produces near-optimal columns – if you boost a better CSSP-algorithm, you get better results. Further, by comparing the performance on Synthetic 1 with Synthetic 2, we see that our algorithm (as well as prior algorithms) gain significantly in performance for rapidly decaying singular values; our new theoretical analysis reflects this behavior, whereas prior results do not.

The theory bound depends on $\epsilon = c/k$. The figure to the right shows a result for $k = 10$; $c = 2k$ ($k$ increases but $\epsilon$ is constant). Comparing the figure with the HGDP plot in Figure 2, we see that the quantitative performance is approximately the same, as the theory predicts (since $c/k$ has not changed). The percentage error stays the same even though we are sampling *more* columns because the benchmark $\|\mathbf{A} - \mathbf{A}_k\|_F$ also get smaller when $k$ increases. Since **ADP-Nopt** is the superior algorithm, we continue with results only for this algorithm.

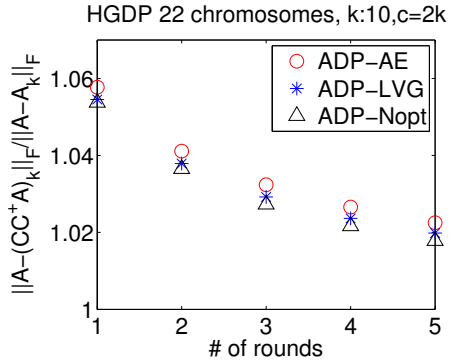

Our next experiment is to test which adaptive strategy works better in practice given the same initial selection of columns. That is, in Figure 2, **ADP-AE** uses an adaptive sampling based on the residual $\mathbf{A} - \mathbf{C}\mathbf{C}^+\mathbf{A}$ and then adaptively samples according to the adaptive strategy in [8]; the initial columns are chosen with the additive error algorithm. Our approach chooses initial columns with the relative error CSSP-algorithm and then continues to sample adaptively based on the relative error CSSP-algorithm and the residual $\mathbf{A} - (\mathbf{C}\mathbf{C}^+\mathbf{A})_{tk}$. We now give all the adaptive sampling algorithms the benefit of the near-optimal initial columns chosen in the first round by the al-

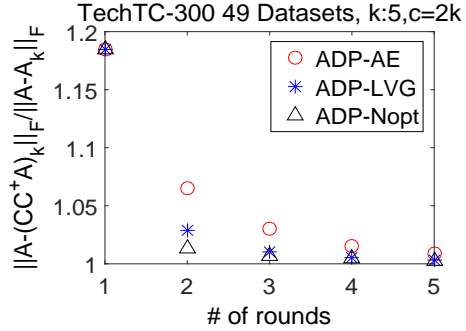

gorithm from [15]. The result shown to the right confirms that **ADP-Nopt** is best even if all adaptive strategies *start from the same initial near-optimal columns*.

**Adaptive versus Continued Sequential Sampling.** Our last experiment is to demonstrate that adaptive sampling works better than continued sequential sampling. We consider the relative error CSSP-algorithm in [15] in two modes. The first is **ADP-Nopt**, which is our adaptive sampling algorithms which selects $tc$ columns in $t$ rounds of $c$ columns each. The second is **SEQ-Nopt**, which is just the relative error CSSP-algorithm in [15] sampling $tc$ columns, all in one go. The results are shown on the right. The adaptive boosting of the relative error CSSP-algorithm can gives up to a 1% improvement in this data set.

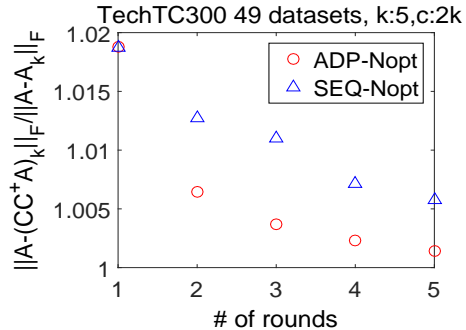

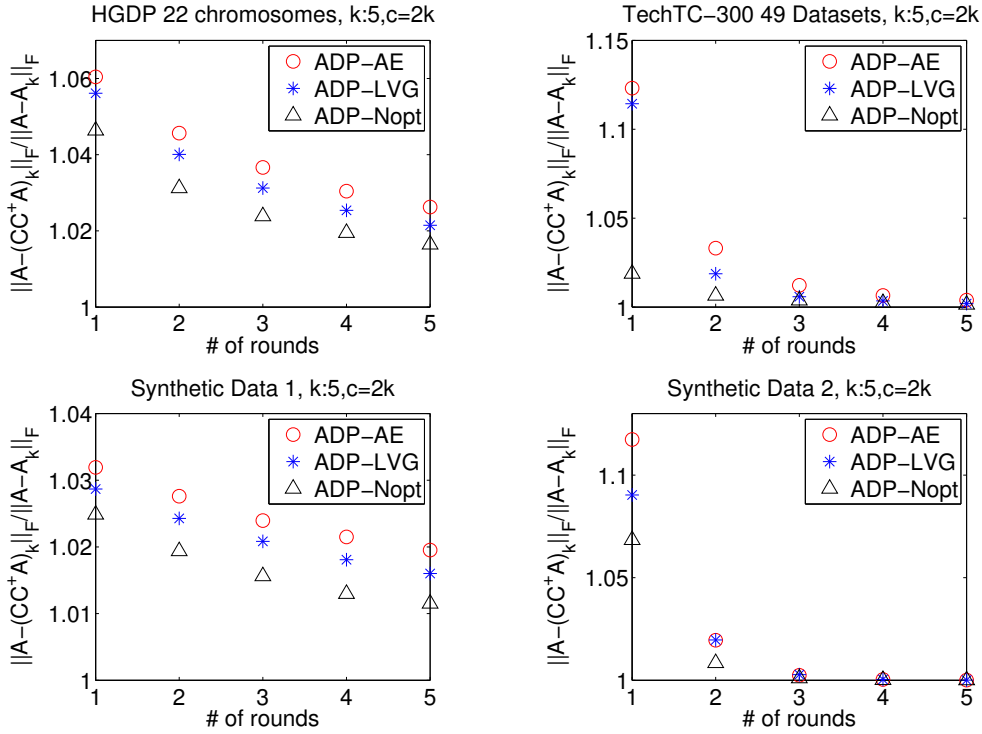

Figure 2: Plots of relative error ratio $\left\| \mathbf{A} - (\mathbf{C}\mathbf{C}^{+}\mathbf{A})_k \right\|_F / \left\| \mathbf{A} - \mathbf{A}_k \right\|_F$ for various adaptive sampling algorithms for $k = 5$ and $c = 2k$. In all cases, performance improves with more rounds of sampling, and rapidly converges to a relative reconstruction error of 1. This is most so in data matrices with singular values that decay quickly (such as TectTC and Synthetic 2). The HGDP singular values decay slowly because missing entries are selected randomly, and Synthetic 1 has slowly decaying power-law singular values by construction.

## 4 Conclusion

We present a new approach for adaptive sampling algorithms which can boost relative error CSSP-algorithms, in particular the near optimal CSSP-algorithm in [15]. We showed theoretical and experimental evidence that our new adaptively boosted CSSP-algorithm is better than the prior existing adaptive sampling algorithm which is based on the additive error CSSP-algorithm in [11]. We also showed evidence (theoretical and empirical) that our adaptive sampling algorithms are better than sequentially sampling all the columns at once. In particular, our theoretical bounds give a result which is tighter for matrices whose singular values decay rapidly.

Several interesting questions remain. We showed that the simplest adaptive sampling algorithm which samples a constant number of columns in each round improves upon sequential sampling all at once. What is the optimal sampling schedule, and does it depend on the singular value spectrum of the data matric? In particular, can improved theoretical bounds or empirical performance be obtained by carefully choosing how many columns to select in each round?

It would also be interesting to see the improved adaptive sampling boosting of CSSP-algorithms in the actual applications which require column selection (such as sparse PCA or unsupervised feature selection). How do the improved theoretical estimates we have derived carry over to these problems (theoretically or empirically)? We leave these directions for future work.

**Acknowledgements**
Most of the work was done when SP was a graduate student at RPI. PD was supported by IIS-1447283 and IIS-1319280.

## Footnotes

[1] $\mathbf{C}\mathbf{C}^+\mathbf{A}$ is the best possible reconstruction of $\mathbf{A}$ by projection into the space spanned by the columns of $\mathbf{C}$.

[2]For an additive-error CSSP algorithm, $\mathbb{E}_{\mathbf{C}} \left[ \left\| \mathbf{X} - (\mathbf{C}\mathbf{C}^+\mathbf{X})_k \right\|_F^2 \right] \leq \left\| \mathbf{X} - \mathbf{X}_k \right\|_F^2 + \epsilon(c, k) \|\mathbf{X}\|_F^2.$

[3]**ADP-Nopt:** has two stages. The first stage is a deterministic dual set spectral-Frobenius column selection in which ties could occur. We break ties in favor of the column not already selected with the maximum norm.

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
