[Reviews · NeurIPS 2015]

Submitted by Assigned_Reviewer_1

Summary:

This paper provides a new algorithm for column selection - i.e. selecting a small number of columns of a matrix. The paper proposes a small+subtle change in an existing adaptive sampling technique, to obtain better approximation guarantees (in a scaling in epsilon sense).

Quality:

The paper is clear and precise. The algorithm is a small but subtle modification of an existing algorithm. The empirical results are interesting, but do not make a strong case for the method (improvements seem to be somewhat marginal). Proof techniques are somewhat standard. Overall, a well-written paper on a somewhat incremental algorithmic advance.

Clarity:

The paper is quite clearly written, and pretty easy to read and understand. This is its biggest strength.

Originality:

It is not a-priori obvious that the modification proposed here leads to improvements in scaling; hence this paper does have some intellectual merit. However, on the other hand, it is not a completely counter-intuitive modification. So overall the level of originality in the algorithm design is not too high. Ditto for the analysis; again, to too many new techniques as far as this reviewer could see (definitely none highlighted).

Significance:

Overall, the reviewer believes the significance of this work is low. Column sampling is a somewhat niche area within machine learning, and the improvements herein, while analytically interesting, do not seem to yield large improvements in approximation and are also not computationally faster to achieve.
Summary: The paper provides a new algorithm for adaptive column sampling. Analytically, it shows better scaling dependence on the error required; however the empirical results are somewhat underwhelming. Overall, a paper of some interest to a small subset of the NIPS community.

Submitted by Assigned_Reviewer_2

In many applications, the goal is to select a subset of columns from a matrix optimally so that the projection of the matrix onto the columns has minimal reconstruction error. There are two ways to do this: (1) continued sampling, in which the columns must all be selected at once, and (2) adaptive sampling, in which you can first select a few columns, look at the result, and use this information to decide which columns to select next. Adaptive sampling has been analyzed for additive error column selection algorithms but not for relative error column selection algorithms. This paper proposes an adaptive sampling method to use with relative error column selection algorithms.

The main contribution of this paper is theoretical, rather than practical. It seems to resolve an open theoretical question regarding adaptive vs. continued sampling. The paper does not address applications in any serious way, and quite frankly, I'm not sure why one would want to restrict oneself to matrix approximations that are projections onto a subset of columns. It seems that the SVD makes much more sense here....

Editorial note: Please define an additive error CSSP algorithm! You do it for relative error CSSP, but not additive error.
Summary: This paper seems like a solid contribution to the problem of optimally selecting columns from a matrix.

Submitted by Assigned_Reviewer_3

The paper studies the column subset selection problem (CSSP) via adaptive sampling. The mains points are:

1) The adaptive sampling strategy developed in [8] achieves better reconstruction error than continued sampling for additive error CSSP-algorithms, but gives a worse reconstruction error guarantee than continued sampling for relative error CSSP-algorithms.

2) The paper proposes a novel adaptive sampling strategy which is a subtle modification to the strategy presented in [8]. Instead of subtracting the full projection for computing the residual matrix, the proposed strategy in the l-th sampling round subtracts the best rank-lk approximation to the target matrix in the span of the columns selected from the first l-1 rounds. The theorem shows that the proposed strategy is able to achieve better theoretical guarantee over continued sampling for relative-error CSSP algorithms.

3) The authors implemented the proposed strategy using two relative-error column selection algorithms. Experiments show that they outperform continued sampling and prior adaptive sampling approaches.

Overall, I like this paper. It is well written and is a pleasure to read. Although the approach introduced in this paper is incremental to the existing literature, it is solid and technically sound. To me, this paper asks an interesting question and addresses it in a simple but elegant way. I think it should be of interest to a subset of the NIPS audience.

On the negative side, the claim that data matrices have rapidly decaying singular values is not always true. For example, Bai-Yin's law points out that if the entries of a m*n (m >= n) matrix A are independent standard normal random variables, then almost surely the largest singular value is around sqrt(m)+sqrt(n), while the smallest singular value is around sqrt(m)-sqrt(n). This suggests that all the singular values of A are pretty similar if A is a tall-and-skinny matrix. Since the theoretical guarantee of this paper heavily depends on the rapid decay of the spectrum of the target matrix, I am afraid that the proposed approach will perform poorly when dealing with matrix with almost flat singular values. I would have liked to see some discussions about this issue.

Minor issues:

- Line 74: is t rounds -> in t rounds - Line 411: in[15] -> in [15] - In page 7, the font size in figures are not consistent.
Summary: The paper presents an interesting adaptive sampling approach for the column subset selection problem. It reports both theoretical and empirical evidence that the proposed approach is better than non-adaptive sampling and prior adaptive sampling approaches. However, its assumption about rapidly decaying singular values may prevent its application to some problems. I am hesitating between a 6 and 7.

Author Feedback
Author rebuttal: Reviewer_1:
Originality: The modification proposed in our paper is subtle, but significant. We are the first to achieve a tighter theoretical bound on the approximation error. This has remained an open problem for over 10 years. Our result is never worse than the existing, but gives significantly better performance with decaying spectrum.

Significance: Column sampling has wide use in machine learning, like Nystrom approximation, CUR decomposition, core-set construction for least squares, and has been used to derive provably accurate feature selection methods for k-means, SVM, etc.

Reviewer_2:
We thank the reviewer for the review. We will add the additive-error CSSP algorithm in the final version. Thanks for pointing this out.

Why column sampling instead of SVD ? Motivated by the fact that in many practical applications, the left and right singular vectors of a matrix A lacks any physical interpretation, a long line of work (see References 6,7,8,9,10,11,12,13,14 in our paper) focussed on extracting a subset of columns of the matrix A, which are approx as good as Ak at reconstructing A.

In the final version we can make this motivation more explicit to the readers.

Reviewer_3:
We will fix the typos and the fonts in the figure in the final version. Thanks for pointing this out.

"On the negative side...."
The reviewer points out that in real data, the spectrum does not decay for tall thin matrices, which is not a relevant application for column selection. The realm of application of column-subset selection is when k is small compared to the true rank of the matrix, in which case we showed data matrices with quickly decaying rank. Random matrices are rare in practice, which is the counter-example the reviewer gave; nevertheless, even for random matrices, our result is better than previous existing results.

Reviewer_5:
We thank the reviewer for the comments. A detailed comparison of existing bounds and algorithms was not possible within the page limit. We will address these issues in a full version of the paper.

Reviewer_6:
Here are answers to the reviewer's questions
1. All the column subset algorithms in the comparison sample r=ct columns. Our algorithm breaks this down to t adaptive rounds of c columns each.
2. As mentioned in Comment 2 after Theorem 1, many CSSP algorithms deliver a result in expectation as in [15]. The BSS algorithm used in [15] is actually deterministic. For the high probability CSSP algorithm, each round of adaptive sampling can introduce additional probability of failure. However, you can always boost this to high probability by repeating the experiment and choosing the best set of ct columns. Therefore, t does not play a significant role in the probability of failure -- it only affects the running time of the algorithm. t is a parameter that can be chosen to optimize the bound in the theorem. We have not fully explored this dimension, and our results focus on small t.